# Variant-specific introduction and dispersal dynamics of SARS-CoV-2 in New York City – from Alpha to Omicron

Simon Dellicour[1,2]*, Samuel L. Hong[2], Verity Hill[3], Dacia Dimartino[4], Christian Marier[4], Paul Zappile[4], Gordon W. Harkins[5], Philippe Lemey[2], Guy Baele[2], Ralf Duerr[6,7,8]*, Adriana Heguy[4,9]*

1 Spatial Epidemiology Lab (SpELL), Université Libre de Bruxelles, Bruxelles, Belgium, 2 Department of Microbiology, Immunology and Transplantation, Rega Institute, Laboratory for Clinical and Epidemiological Virology, KU Leuven, Leuven, Belgium, 3 Department of Epidemiology of Microbial Diseases, Yale School of Public Health, New Haven, Connecticut, United States of America, 4 Genome Technology Center, Office for Science and Research, NYU Langone Health, New York, New York, United States of America, 5 South African National Bioinformatics Institute, University of the Western Cape, Bellville, South Africa, 6 Department of Microbiology, NYU Grossman School of Medicine, New York, New York, United States of America, 7 Department of Medicine, NYU Grossman School of Medicine, New York, New York, United States of America, 8 Vaccine Center, NYU Grossman School of Medicine, New York, New York, United States of America, 9 Department of Pathology, NYU Grossman School of Medicine, New York, New York, United States of America

* simon.dellicour@ulb.be (SD); ralf.duerr@nyulangone.org (RD); adriana.heguy@nyulangone.org (AH)

**Data Availability Statement:** R scripts and related files needed to run all the phylogeographic analyses, as well as BEAST XML files, are all available at https://github.com/sdellicour/new_york_variants.

## Abstract

Since the latter part of 2020, SARS-CoV-2 evolution has been characterised by the emergence of viral variants associated with distinct biological characteristics. While the main research focus has centred on the ability of new variants to increase in frequency and impact the effective reproductive number of the virus, less attention has been placed on their relative ability to establish transmission chains and to spread through a geographic area. Here, we describe a phylogeographic approach to estimate and compare the introduction and dispersal dynamics of the main SARS-CoV-2 variants – Alpha, Iota, Delta, and Omicron – that circulated in the New York City area between 2020 and 2022. Notably, our results indicate that Delta had a lower ability to establish sustained transmission chains in the NYC area and that Omicron (BA.1) was the variant fastest to disseminate across the study area. The analytical approach presented here complements non-spatially-explicit analytical approaches that seek a better understanding of the epidemiological differences that exist among successive SARS-CoV-2 variants of concern.

## Author summary

The evolution of SARS-CoV-2, the virus responsible for the coronavirus disease 2019 (COVID-19) pandemic, has seen the emergence of novel variants with increased transmissibility, virulence and/or ability to escape the immunity induced by previous infections and vaccination. In our study, we determined to what extent these viral variants differed

**Funding:** SD acknowledges support from the *Fonds National de la Recherche Scientifique* (F.R. S.-FNRS, Belgium; grant n˚F.4515.22). SD and PL acknowledge support from the European Union Horizon 2020 project MOOD (grant agreement n˚ 874850). SD and GB acknowledge support from the Research Foundation - Flanders (*Fonds voor Wetenschappelijk Onderzoek - Vlaanderen*, FWO, Belgium; grant n°G098321N). PL acknowledges support from the European Research Council under the European Union's Horizon 2020 research and innovation programme (grant agreement n˚ 725422 - ReservoirDOCS), the Wellcome Trust through project 206298/Z/17/Z, and the National Institutes of Health grant R01 AI153044. SLH and GB acknowledge support from the Research Foundation - Flanders (*Fonds voor Wetenschappelijk Onderzoek-Vlaanderen*, FWO, Belgium; grant n°G0E1420N). GWH acknowledges support from The National Institutes of Health, USA (grant n°5U01AI152151-03). GB acknowledges support from the Internal Funds KU Leuven (grant n°C14/18/094). The funders had no role in study design, data collection and analysis, decision to publish, or preparation of the manuscript.

**Competing interests:** The authors have declared that no competing interests exist.

in their dispersal dynamics at a local scale. Specifically, we analysed viral genomes to reconstruct, in space and time, the dispersal history of specific variants to subsequently compare their ability to establish local transmission chains and spread through the same study area. Our study focuses on the New York City area, which has been associated with high genomic surveillance efforts. We here took advantage of the resulting genomic data set to compare and highlight notable differences in the introduction and dispersal dynamics of the Alpha, Iota, Delta, and Omicron variants in the New York City area.

## Introduction

Severe acute respiratory syndrome coronavirus 2 (SARS-CoV-2) evolution has been characterised by the emergence of sets of mutations impacting its biological characteristics, likely in response to the changing profile of immunity within the population [1]. In late 2020, viral lineages associated with spike protein mutations such as N501Y and/or E484K, began to be classified as 'variants of interest' (VOIs) or 'variants of concern' (VOCs) by public health authorities. To monitor the emergence and spread of VOIs and VOCs, many countries began to implement systematic SARS-CoV-2 genomic surveillance efforts [2,3]. The World Health Organisation (WHO) defined a VOI as a SARS-CoV-2 variant involving genetic changes that are predicted or known to affect the virus' characteristics, and a VOC as a SARS-CoV-2 variant that meets the definition of a VOI and that has been demonstrated to be associated with one or more of the following changes: increase in transmissibility, increase in virulence, or decrease in effectiveness of public health/social measures or available diagnostics, vaccines, therapeutics (see the dedicated WHO webpage for the detailed definition; http://www.who.int/activities/tracking-SARS-CoV-2-variants; accessed June 30, 2022). To date, five SARS-CoV-2 variants have been classified as VOCs: the Alpha variant (lineage B.1.1.7) first detected in late 2020 in England, the Beta variant (lineage B.1.351) first detected in late 2020 in South Africa, the Gamma variant (lineage P.1) first detected in early 2021 in Brazil, the Delta variant (lineage B.1.617.2) first detected in late 2020 in India, and the Omicron variant (children of lineage B.1.1.529, including lineages BA.*) first detected in late 2021 in Botswana and South Africa [4–7].

Because VOCs have been responsible for several epidemic surges worldwide, their ability to grow in frequency and increase the effective reproductive number of the virus has been studied extensively in different countries [7–9]. However, whether differences exist in their relative ability to establish sustained transmission chains into a geographic area and to further disperse through such an area remains largely unknown. The objective of the present study is to compare the introduction and dispersal dynamics of the main VOI/VOCs that have spread within New York City (NYC) and its surrounding counties in New York State (Nassau, Suffolk, and Westchester), hereafter referred to as the 'NYC area'. Through our multi-center healthcare institution at New York University (NYU), comprehensive genomic surveillance has been conducted consistently since the beginning of the pandemic in the study area.

The first positive case of SARS-CoV-2 in NYC was identified on February 29, 2020, soon after followed by the detection of community transmission and NYC being designated as the epicentre of the COVID-19 epidemic in the United States [10–12]. As such the NYC area was severely impacted during the first phase of the SARS-CoV-2 pandemic, which was largely attributed to NYC being a global hub, but also to the delayed recognition of SARS-CoV-2 transmission in the city [13]. When comparing the five NYC boroughs, substantial differences in SARS-CoV-2 transmission rates, dispersal dynamics, and hospitalizations were highlighted [12,14]. In late 2020/early 2021, the first VOIs/VOCs started circulating in New York City/

State (see Fig 1 for the evolution of the estimated relative abundance of the main VOI/VOCs across New York State). VOI Iota apparently arose in NYC and caused a regional epidemic, which also coincided with the global spread and introduction of VOC Alpha into the NYC area [15–18]. In early 2021, VOC Gamma was also introduced into the NYC area [19], but its circulation remained relatively limited compared to the other variants (Fig 1). In times of increasing vaccine coverage and decreasing levels of Alpha and Iota infections in New York (May/June 2021), a new wave was ignited by the VOC Delta and its various AY.* sublineages, which dominated the second half of 2021 [20]. In late 2021/early 2022, the waning Delta variant co-circulated for a few months with the rapidly emerging VOC Omicron (Fig 1), first detected in Sub-Saharan countries, including South Africa [20,21]. Overall, Omicron and its BA.* sub-variants have predominated in New York State in 2022.

Here, we build on a previous analytical pipeline [22] to implement a phylogeographic approach and introduce metrics to compare the introduction and dispersal dynamics of the main VOI/VOCs that spread across the NYC area, namely Iota, Alpha, Delta, and Omicron (BA.1). For this purpose, we exploit a comprehensive data set of SARS-CoV-2 sequences generated by a genomic surveillance effort in a large metropolitan healthcare system with hospitals in several city boroughs and adjacent suburban areas. For each variant under consideration, the resulting data set consists of all sequences available from the study area as well as a set of 'background' sequences from surrounding US states, countries and other parts of the world, and which was used to delineate the phylogenetic clades corresponding to distinct introduction events in the study area. Our analyses allowed us to compare the abilities of the different variants to be introduced and further disperse within the study area, and revealed that although Delta had the highest number of detected introduction events into the NYC area, it had a lower ability to establish sustained transmission chains throughout this area. This differs for Omicron (BA.1) that had both the second highest number of detected introductions and was the variant fastest to disseminate across the area.

## Results

### Identifying distinct introduction events into the study area

For each variant, we first performed a preliminary discrete phylogeographic analysis [24] to identify clades (including clades of size $n = 1$ sequence) that likely arose from distinct introduction events into the NYC area. Each of those clades, or 'clusters', thus corresponds to a sample from a local transmission chain that had a distinct entry point in the study area. We identified 319 introduction events for Iota (95% highest posterior density interval [HPD] = 304–331), 780 for Alpha (95% HPD = 752–817), 3653 for Delta (95% HPD = 3622–3691), and 2253 for Omicron (BA.1; 95% HPD = 2225–2273). For all those variants, the total number of introduced clades contained a relatively high proportion that comprised only a single sequence. This proportion was 65.6% for Iota (95% HPD = 63.6–68.5), 66.1% for Alpha (95% HPD = 64.7–67.3), 83.9% for Delta (95% HPD = 75.7–76.9), and 78.8% for Omicron (BA.1; 95% HPD = 78.0–79.6). Although the sampling and identification of a single sequence per clade does in no way prove the absence of further transmission following the introduction event, it likely indicates limited onward circulation. Of note, we identified one particularly large clade containing most of the Iota sequences (2040 out of 2519 sequences collected within the NYC area), which is in line with the findings of previous phylogeographic analyses that inferred a New York origin for that VOI [15]. Consequently, all the other Iota clades would then correspond to re-introduction events into the study area.

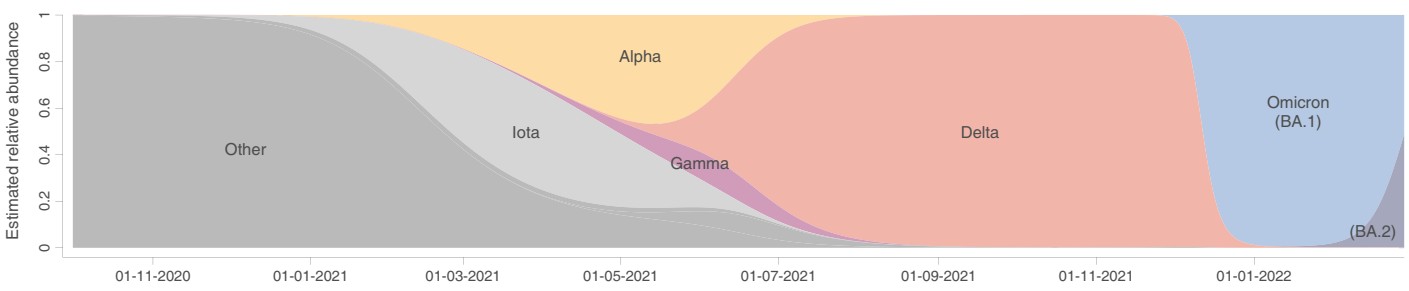

**Fig 1. Evolution of the relative abundance of main SARS-CoV-2 VOI/VOCs in New York State.** Source of data: GISAID (www.gisaid.org).

### Reconstructing the local dispersal history of viral lineages

We then performed discrete [24] and continuous [25,26] phylogeographic reconstructions along the NYC area clades identified as introductions in the previous step. These phylogeographic analyses permitted the reconstruction of the local dispersal history of sampled viral lineages across the study area, as well as subsequent quantitative comparisons of the introduction and dispersal capacities of each variant. An initial visual comparison of our discrete phylogeographic reconstructions for the four variants under consideration already revealed distinct variant-specific dispersal patterns for the viral lineages (Fig 2). For example, whereas Iota and Omicron (BA.1) exhibited numerous supported transition events among non-adjacent boroughs/counties, this was not the case for the Alpha variant where most supported lineage transition events are inferred to have occurred among neighbouring boroughs/counties. In contrast to the other variants, the discrete phylogeographic reconstruction inferred for the Delta variant identified a restricted number of supported transition events and hardly any connections with other boroughs/counties in the study area.

### Comparing the introduction patterns of SARS-CoV-2 variants

To compare the introduction patterns among variants, we defined and estimated two different metrics that aimed at quantifying the capacity of each variant to establish local transmission chains: (i) the probability $p_1$ that two circulating lineages drawn at random belong to the same clade/cluster introduced into the study area, and (ii) the proportion $p_2$ computed as the ratio between the number of circulating clusters and the number of phylogenetic branches occurring at the same time across the study area. We here define a "circulating cluster" as a clade introduced into the study area and connecting at least three sampled viral genomes. The estimation of those metrics were both based on the discrete phylogeographic reconstructions performed along NYC area clades containing at least three genomes sampled in the study area.

The evolution of $p_1$ and $p_2$ through time clearly confirms that, because of its likely emergence in the study area, Iota exhibits a strikingly different introduction pattern relative to the other variants. First, the probability $p_1$ that two circulating Iota lineages belong to the same cluster was close to '1' during the early months after its emergence (solid grey curve; Fig 3A). This probability $p_1$ tended to increase while remaining relatively low compared to the other variants, which would then correspond to the contribution of Iota re-introduction events into the study area. Second, the number of circulating clusters (relative to the number of circulating phylogeny branches) tended to remain much lower for Iota compared to the other variants (dashed curves; Fig 3A).

Alpha, Delta, and Omicron (BA.1) exhibited more similar patterns of introduction as estimated by $p_1$ and $p_2$, however differences among them existed. For example, Delta was

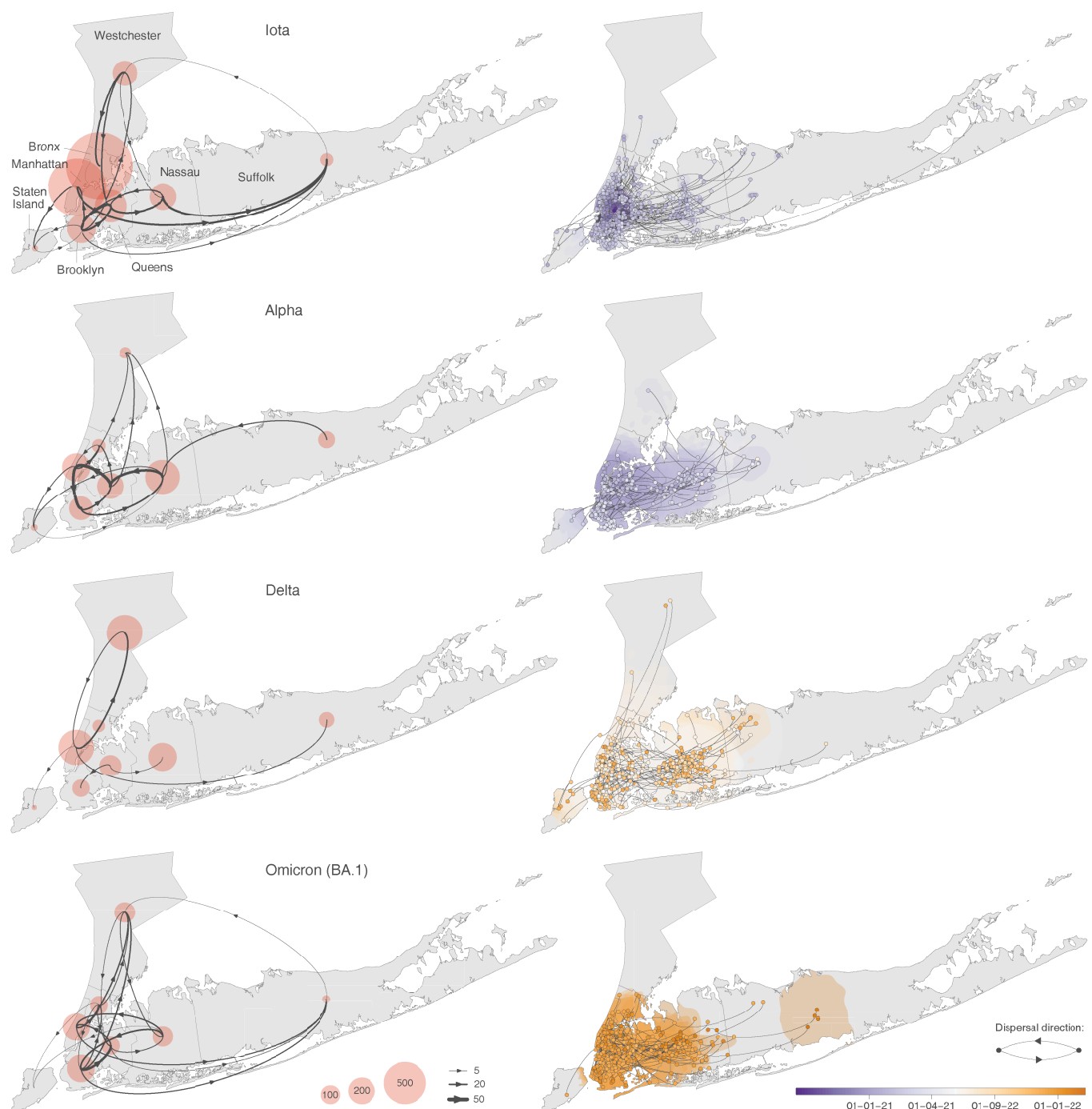

**Fig 2. Investigating the dispersal patterns of sampled SARS-CoV-2 lineages in the NYC area.** We here report both a discrete (left) and continuous (right) phylogeographic reconstruction of the dispersal history of viral lineages belonging to the Iota, Alpha, Delta, and Omicron (BA.1) variants. For the discrete reconstructions, we report the number of lineage dispersal events inferred between (arrows) and within (transparent red circles) boroughs/counties of the NYC area, both measures being averaged over 900 trees sampled from each posterior distribution. Specifically, we only report averaged numbers of lineage dispersal events between boroughs/counties associated with an adjusted Bayes factor support higher than 3, which corresponds to a 'positive support' [23]. For the continuous reconstructions, we map the maximum clade credibility (MCC) tree and overall 80% highest posterior density (HPD) regions reflecting the uncertainty related to the Bayesian phylogeographic inference. MCC trees and 80% HPD regions are based on 900 trees sampled from each posterior distribution. MCC tree nodes are coloured according to their time of occurrence, and 80% HPD regions were computed for successive time layers and then superimposed using the same colour scale reflecting time. The underlying map has been retrieved from the Database of Global Administrative Areas (GADM; https://gadm.org).

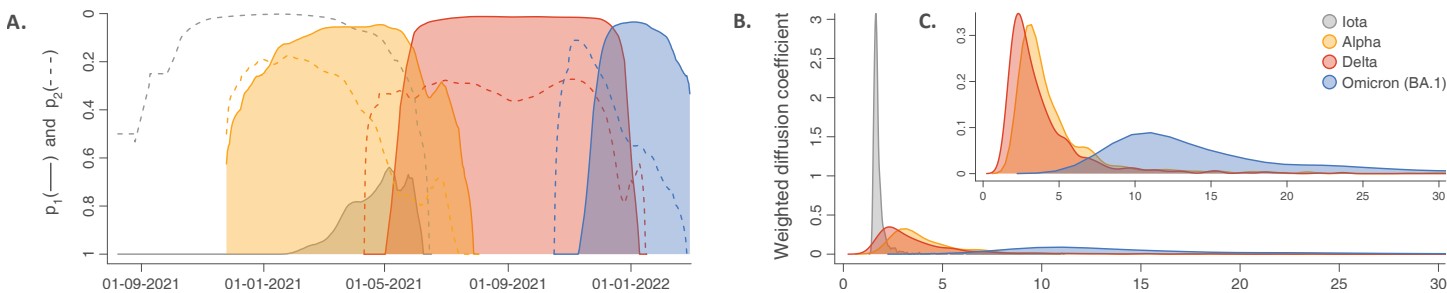

**Fig 3. Comparing the introduction and dispersal dynamics of SARS-CoV-2 variants in the NYC area. A:** Evolution of (i) the probability $p_1$ that two circulating lineages drawn at random belong to the same clade/cluster introduced into the study area (solid curves), and (ii) the proportion $p_2$ computed as the ratio between the number of circulating clusters and the number of phylogenetic branches occurring at the same time across the study area (dashed curves). **B:** Posterior distributions of the weighted diffusion coefficient ($km^2$/day) estimated for each variant. **C:** Posterior distributions of the weighted diffusion coefficient estimated for each variant except Iota, which allows a focus on the results obtained for the three other variants. All the results reported in Fig 3 are based on 900 trees sampled from each posterior distribution obtained by discrete (A) or continuous (B-C) phylogeographic inference.

associated with a relatively long period of time when the probability $p_1$ of sampling two Delta lineages from the same introduction event was close to zero which was not mirrored in Alpha or BA.1 (solid curves; Fig 3A). Similarly, maximal $p_2$ estimates for Delta were lower than the $p_2$ maxima estimated for Alpha and BA.1, which would further reflect a slightly lower propensity to establish large clusters in the study area.

## Comparing the dispersal dynamics of SARS-CoV-2 variants

To compare the dispersal dynamics among the variants considered, we proposed and estimated two different metrics that quantify the capacity of a distinct introduction event to spread across the study area. The first metric, $p_3$, was defined as the proportion of phylogeny branches associated with a transition event among boroughs/counties and was estimated from our discrete phylogeographic reconstructions. This metric was averaged among clusters and specifically allowed investigation of the extent to which distinct independent introduction events managed to spread among given administrative units (here US counties). This time estimated from our discrete phylogeographic reconstructions, the second metric was the weighted diffusion coefficient introduced by Trovão and colleagues [27], which is a conceptually different dispersal metric that quantifies the velocity at which lineages diffused within the study area and was estimated from the continuous phylogeographic analyses.

For the metric $p_3$, we estimated a value of 0.21 for Iota (95% HPD = 0.20–0.23), 0.24 for Alpha (95% HPD = 0.23–0.25), 0.18 for Delta (95% HPD = 0.18–0.19), and 0.29 for BA.1 (95% HPD = 0.28–0.30). Thus, according to this metric, Omicron (BA.1), once introduced, demonstrated the highest capacity to spread across different boroughs/counties, followed by Alpha, Iota, and Delta. The estimates of the weighted diffusion coefficient revealed that Omicron (BA.1) was associated with a notably higher diffusion velocity compared to the three other variants (Fig 3B and 3C), implying that Omicron (BA.1) were associated with a higher diffusivity.

## Assessing the sensitivity of the new metrics to different scenarios

Besides the weighted diffusion coefficient that was previously introduced and applied [27], the present study thus describes three other metrics $p_1$-$p_2$-$p_3$ to characterise and compare the introduction/dispersal dynamics of (viral) lineages in a study area. To assess the sensitivity of those metrics to different introduction and dispersal scenarios, we simulated the invasion of the Alpha variant while testing (i) different values for the minimum size of the introduced clusters circulating in the study area (Fig 4A and 4B), (ii) different numbers of introduced clusters

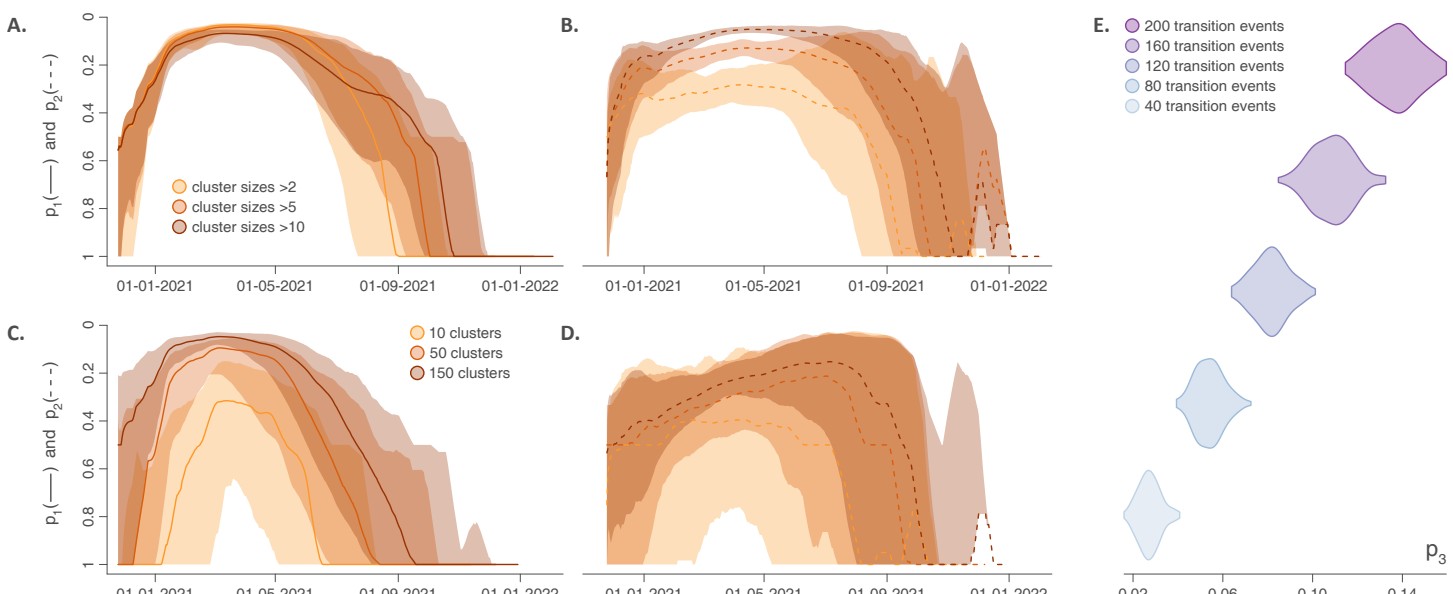

**Fig 4. Sensitivity of the new metrics to different introduction and dispersal scenarios simulated for the Alpha variant. A-B:** Impact of the size of the clusters introduced into the study area on the metrics $p_1$ and $p_2$. **C-D:** Impact of the number of clusters introduced into the study area on the metrics $p_1$ and $p_2$. In panels A-D, mean $p_1$-$p_2$ values and associated 95% HPD intervals are reported by solid/dashed curves and surrounding shaded polygons, respectively. **E:** Impact of the total number of transition events (among counties) on the $p_3$ metric.

(Fig 4C and 4D), and (iii) different total numbers of transition events between counties for that variant (Fig 4E; see the Materials & Methods section for further detail on the simulation processes).

The minimum size of the introduced clusters had a negligible impact on the $p_1$ metric (Fig 4A): around the peak of the epidemic wave, the probability $p_1$ that two circulating lineages drawn at random belong to the same cluster tends to zero in all cases. On the contrary, varying the same simulation parameter had a notable impact on the $p_2$ metric (Fig 4B): the ratio $p_2$ between the numbers of co-occurring clusters and phylogenetic branches clearly tends to increase with the minimum introduced cluster size considered in the simulations. The number of simulated clusters introduced in the study area influenced both $p_1$ (Fig 4C) and $p_2$ (Fig 4D) metrics. In particular, drastically decreasing the number of introduced clusters provokes a shift of the $p_1$ curve, making it closer to the pattern of the $p_1$ curve reported for the Iota variant (Fig 3A), i.e. the variant that was thus associated with a major clade circulating in the study area. Finally, as expected, our results confirm that the $p_3$ metric is very sensitive to the number of simulated transition events among counties (Fig 4E).

## Discussion

The unprecedented amount of genomic data generated through worldwide genomic surveillance of SARS-CoV-2 has enabled valuable insights into the dispersal dynamics of the virus and its different variants. In that context, numerous phylogeographic investigations have been conducted at global [28–31] and local [32–35] scales. By placing phylogenetic trees in geographic space and time, phylogeographic reconstructions provide information on the mode and pace of viral lineage circulation across spatial scales. For instance, phylogeographic analyses have been conducted to investigate the impact of international travel restrictions [32], the origin of specific variants [5,7], or the contribution of international travel to local epidemics [36].

In the present study, we exploit such phylogeographic reconstructions to compare the capacity of important SARS-CoV-2 variants to enter into and disperse through a geographic area. For this purpose, we propose the estimation of a series of metrics aimed at comparing the ability of the main VOI/VOCs (i) to establish local transmission chains in the area and (ii) to spread through it. We here apply our approach to the NYC area, for which genomic surveillance has been continuously conducted throughout the pandemic. Furthermore, we take advantage of the comprehensive data collection associated with SARS-CoV-2 genomes sequenced at NYU Langone Health (NYULH), metadata on the county or zip code area of origin being required to conduct discrete and continuous phylogeographic analyses, respectively (we refer to the Materials and Methods section for more detail).

Our comparison of the introduction patterns confirms that Iota likely emerged within or in close proximity to the NY state area [18,37], and reveals that Delta had the lowest tendency to establish large transmission chains in the NYC area. Iota is characterised by a main cluster corresponding to its initial emergence, as well as a series of smaller ones that emerged towards the end of its circulation period corresponding to re-introduction events in the NYC area. While we infer a higher absolute number of distinct introduction events for Delta, this VOC was also found to have the highest number of introductions associated with a unique sampled genomic sequence. The probability of sampling two Delta sequences originating from the same introduction was close to zero during almost the entire corresponding epidemic phase across the NYC boroughs and studied US counties.

As for the comparison of the dispersal dynamics, our results reveal that Delta was the variant that dispersed the least among boroughs and counties in the study area, and that BA.1 was the variant associated with the highest diffusion velocity. The lowest rate of transition among boroughs/counties estimated for Delta is readily apparent when comparing the supported lineage transition events inferred by discrete phylogeography (Fig 2). Because we used Bayes factor support adjusted for sampling heterogeneity to filter supported transition events (see the Materials and Methods section), such a limited number of supported transitions could also reflect the difficulty of the underlying discrete diffusion model to retrace the main transition routes when based on an uneven sampling effort among locations. This result is however in line with the relatively lower propensity of Delta to establish sustained transmission chains within the study area, as quantified by $p_1$ and $p_2$ estimates (Fig 3A). While we do not have epidemiological evidence that could explain this different pattern for Delta, one potential explanation could lie in the conjunction of a still high vaccine efficiency against that VOC [38,39] and the large local vaccination coverage (www1.nyc.gov) during the Delta wave. Specifically, protection against symptomatic disease and/or diminution of the infectious duration could have been a factor limiting large transmission chains with Delta. The non-pharmaceutical interventions (NPI) having varied through time, their intensity — something that could e.g. be measured by the stringency index — could also have had an impact on the introduction and dispersal dynamics of the successive variants studied here. This index has tended to decrease over the epidemic waves associated with Alpha, Delta, and BA.1 (https://ourworldindata.org/), which could be in line with the higher diffusion velocity of BA.1 and its higher capacity to spread across different boroughs/counties.

The simulations that we conducted to assess the sensitivity of our metrics to the introduction and dispersal scenarios confirm that those metrics can capture notable variations in the ability of specific variants to establish local transmission chains and further spread across a study area. While we designed these introduction and dispersal metrics for the comparison of variants with different sampling sizes, our study still exhibits a limitation related to the spatially heterogeneous sampling effort within the study area that directly impacts the phylogeographic reconstructions. This is particularly true for the continuous phylogeographic reconstructions

that were only based on genomic sequences for which the zip code area of origin was known. Because zip codes were exclusively available for the samples sequenced at NYULH, the sampling focuses on the four most populous NYC boroughs (Manhattan, Brooklyn, the Bronx, and Queens) and Nassau County where NYULH has a large hospital. Therefore, the resulting phylogeographic reconstructions, and the continuous ones in particular, remain dependent on each sampling pattern resulting from the genomic surveillance effort related to NYULH. While those phylogeographic reconstructions allow estimating the dispersal history of *sampled* lineages, they thus do not necessarily constitute a realistic summary of the total number of transmission chains in the NYC area. In the present study, phylogeographic analyses were employed to analyse and compare the introduction and dispersal dynamics of target variants rather than trying to precisely reconstruct their overall dispersal history inside the NYC area. Furthermore, the same spatially heterogeneous genomic surveillance effort was applied for all four VOI/VOCs analysed and compared in the present study. In summary, while spatial heterogeneity affects the reconstruction of the overall dispersal history of each variant, we argue that it marginally affects the comparison of their introduction and dispersal dynamics.

As detailed in previous work [22], we also acknowledge that an analysis where the phylogenetic tree and ancestral locations are jointly inferred would be preferable to explicitly take into account the uncertainty associated with Bayesian phylogenetic inference. When dealing with alignments gathering tens of thousands of genomic sequences, as is the case in the present study (up to more than 60,000 sequences in the Delta alignment), such fully integrated Bayesian analyses are not feasible in a reasonable amount of time. As an illustration of the computational burden associated with data sets of that size, the maximum-likelihood phylogenetic inference performed in IQTREE for the Delta and Omicron (BA.1) variants have respectively required more than 130 and 195 hours of running time on a high-performance workstation (40-core/64GB RAM Dell Precision). In that context and while it comes with the limitation of not assessing the effect of phylogenetic uncertainty on the outcomes, using an empirical time-scaled phylogenetic tree represents an interesting compromise to run phylogeographic analyses on large-scale data sets. In their study based on much smaller data set of genomic sequences, Dellicour and colleagues assessed the robustness of their results (inferred number of introduction events into the study area, estimates of lineage dispersal velocity) to the selection of the maximum-likelihood starting tree, and confirmed that the different estimates remain robust irrespective of the choice of the starting tree [22].

In conclusion, we present a phylogeographic approach to compare the ability of the main VOI/VOCs to be introduced and diffuse through the NYC area up to February 2022, highlighting notable differences among Iota, Alpha, Delta, and Omicron (BA.1). By complementing non-spatially explicit analytical approaches (e.g. focusing on the impact of variants on the effective reproduction number), the introduction/dispersal metrics described in the present study improve our understanding of the epidemiological differences that exist among successive SARS-CoV-2 variants of concern.

## Material and methods

### SARS-CoV-2 genomic sequencing

The 5,577 new genomic sequences included in this study were obtained from samples collected in the New York University Langone Health (NYULH) system from December 1, 2020, to February 27, 2022. Genomic surveillance was carried out as described previously [16,20], using the IDT XGen (formerly Swift Normalase) SARS-CoV-2 amplicon-based library prep method run on the Illumina NovaSeq 6000 system on SP 300 cycle flow cells. Only SARS-CoV-2 sequences that were >23,000 bp or >4000x genome coverage were considered adequate and

were included in the analyses. Sequencing reads were demultiplexed using the Illumina bcl2fastq2 Conversion software v2.20 and adapters and low-quality bases were trimmed with Trimmomatic v0.36 [40]. The program BWA v0.7.17 [41] was then used for mapping reads to the SARS-CoV-2 reference genome (NC_045512.2, wuhCor1) and duplicate reads were removed using Sambamba v0.6.8 [42]. GATK v3.8 DepthOfCoverage and HaplotypeCaller tools [43] were used to determine on-target viral coverage and call mutations. Finally, we used the program Pango version v.3.1.20 [44] to determine the SARS-CoV-2 lineage of each sample. All generated sequences were deposited on the GISAID database [45].

## Time-scaled phylogenetic inference

For each variant, our time-scaled phylogenetic inference was based on a comprehensive data set of genomic sequences made up of (i) all sequences obtained through the NYULH genomic surveillance effort (see above) as by the end of February 2022, (ii) all sequences collected in the study area and publicly available on GISAID (www.gisaid.org) as by the end of February 2022, as well as (iii) a set of 'background' sequences which were used in the Nextstrain [46] build focused on North America (but that also includes genomic sequences from the other continents). Specifically, for each variant, we used the North American Nextstrain build that was available on the last collection date of the considered variant in our data set. The purpose of the inclusion of such a background data set is to allow placing NYC clades in a broader global context of SARS-CoV-2 phylogenetic diversity. While we cannot exclude that the selection and/or size of the background data set could influence the subsequent delimitation of NYC clades introduced into the study area, the same approach has here been applied to each variant under consideration, which guarantees a form a consistency in the clade delimitation: if an underestimation of the delineated clades applies, it should similarly affect all four variants and still allow further comparisons of their introduction dynamics. The resulting data sets were made up of 11,758 (Iota), 16,395 (Alpha), 60,019 (Delta), and 32,322 (Omicron-BA.1) genomic sequences. For each data set, we mapped the corresponding sequences against a SARS-CoV-2 reference genome (Genbank ID: NC_045512.2) using minimap2 v2.24, trimmed the data to positions 265–29,674 and padded with Ns to mask out 3' and 5' UTRs.

Time-scaled phylogenies were inferring using a two-step procedure consisting of (i) first estimating a maximum-likelihood (ML) phylogeny using IQ-TREE 2.2.0.3 [47] and (ii) subsequently time-calibrating the resulting maximum-likelihood tree using TreeTime 0.7.4 [48], specifying an evolutionary rate of $8 \times 10^{-4}$ substitutions per site per year (s/s/y), as in the Nextstrain workflow. For variants Iota and Alpha, the ML tree was inferred under a general time-reversible (GTR) model of nucleotide substitution with empirical base frequencies and a four-category FreeRate model of site heterogeneity, which was selected as the best-fitting model using IQTREE's ModelTest functionality. Because of the computational demands required to infer trees with a large number of taxa, we performed an additional step to estimate the ML phylogeny for variants Delta and Omicron (BA.1), where we first constructed a parsimony tree using Online matOptimize [49] and used the resulting tree as a starting candidate tree to infer a ML tree IQ-TREE under a HKY85 nucleotide substitution model with empirical base frequencies. For the Delta variant, the tree was inferred without rate heterogeneity in the substitution model due to the computational demands required and, in the case of the Omicron (BA.1) variant, the tree was inferred using gamma-distributed rate variation among sites. The initial parsimony tree was obtained after 4 rounds of optimization using SPR radius values ranging from 10 to 100 using the method described by Thornlow and colleagues [50].

## Identifying the distinct introduction events

We performed preliminary phylogeographic analyses to identify internal nodes and descendent clades that likely correspond to distinct introductions into, and their subsequent spread within, the NYC area [22]. For this purpose, we employed the discrete phylogeographic model [24] implemented in the software package BEAST 1.10 [51], and only considered two discrete locations: 'NYC area' and 'other'. Those analyses were based on the empirical tree topology corresponding to the time-scaled phylogenetic tree obtained at the previous step for each variant. Each Bayesian inference through MCMC was run for $10^5$ iterations and sampled every 1,000 iterations. Convergence and mixing aspects of all relevant parameters were inspected using Tracer 1.7 [52] to ensure that their associated effective sample size (ESS) values were all >200. After having discarded 10% of sampled posterior trees as burn-in, we constructed maximum clade credibility (MCC) trees using TreeAnnotator 1.10 [51]. The resulting MCC tree obtained for each variant was subsequently used to delineate NYC area clades corresponding to independent introduction events in the study area. In practice, we identified introduction events by comparing the locations assigned to each pair of nodes connected by a phylogenetic branch, i.e. the most probable location inferred at internal nodes and the sampling location for tip nodes. We considered an introduction event to have occurred when the location assigned to a node was 'NYC area' and the location assigned to its parent node in the tree was 'other' [22].

## Local phylogeographic reconstructions

To infer the dispersal history of viral lineages across the study area, we subsequently performed discrete and continuous phylogeographic analyses along the NYC area clades delineated above that consisted of at least three NCY area genomes. In order to identify the best-supported lineage transitions events between NYC area boroughs/counties treated as discrete locations, we used the Bayesian stochastic search variable selection (BSSVS) approach [24] implemented in BEAST 1.10. Each MCMC was run for $5 \times 10^8$ iterations and sampled every $5 \times 10^5$ iterations, except the MCMC analysis dedicated to the Iota variant which was run for $10^9$ iterations while sampling every $10^6$ iterations. MCMC convergence and mixing properties were inspected with Tracer as described above. Statistical support associated with lineage transition events connecting each pair of boroughs/counties were obtained by computing adjusted Bayes factors (BFs), that is, BFs that consider the relative abundance of samples by location [12,53]. Based on a methodology similar to the tip-date randomisation test for temporal signal [54], the adjusted BF takes into account the relative abundance of samples by location [53]. These discrete phylogeographic inferences were based on the following numbers of NYC samples: $n$ = 2182 for Iota, 849 for Alpha, 1015 for Delta, and 728 for Omicron (BA.1).

We used the relaxed random walk (RRW) diffusion model [25,26] implemented in BEAST 1.10 to perform the continuous phylogeographic reconstructions. Continuous phylogeographic inference requires unique sampling coordinates assigned to the tips of the tree. For each sampled genome, we retrieved geographic coordinates from a point randomly sampled within its zip code area of origin, which is the highest level of spatial precision in the available NYULH metadata [12]. Sampled genomes for which the zip code area of origin was unknown or unavailable were discarded from the analyses, which resulted in the following numbers of samples being discarded: $n$ = 344 for Iota, 140 for Alpha, 275 for Delta, and 245 for Omicron (BA.1). Each MCMC was run for $10^8$ iterations and sampled every $10^5$ iterations, and we again used the program Tracer to inspect MCMC convergence and mixing properties, as well as TreeAnnotator to identify and annotate MCC trees. Finally, we used functions available in the R package 'seraphim' [55] to extract spatio-temporal information embedded within posterior

trees and visualise the continuous phylogeographic reconstructions. We also used the 'spread-Statistics' function of the R package 'seraphim' to estimate the weighted diffusion coefficient [27]:

$$D_{weightred} = \left( \sum_i d_i^4 \right) / \left( \sum_i t_i^2 \right)$$

where $d_i$ and $t_i$ are, respectively, the geographic (great-circle) distance and the time elapsed on each phylogeny branch.

## Introduction and dispersal simulations

The sensitivity of the metrics $p_1$-$p_2$-$p_3$ to different introduction and dispersal scenarios was assessed by computing those metrics on simulations of the Alpha variant invasion according to scenarios involving different introduction and dispersal dynamics of viral lineages. Specifically, we performed three distinct analyses each based on a set of 100 simulations: a first set to investigate the impact of the size of the clusters introduced into the study area on the metrics $p_1$ and $p_2$, a second one for the impact of the number of clusters introduced into the study area on the same two metrics, and a third set of simulations to test the impact of the number of transition events on the $p_3$ metric. For the first analysis, simulations were performed by randomly replacing the introduced Alpha clusters by Alpha/Delta/BA.1 clusters sampled with replacement among the overall set of Alpha/Delta/BA.1 clusters associated with a specific minimum size (>2, >5, and >10 sampled sequences). For the second analysis, 100 simulations were performed by randomly resampling different numbers of Alpha variant clusters (10, 50, and 150 clusters). For this second analysis, the timing of introduction of each resampled Alpha cluster was randomly defined by drawing a date from the distribution of inferred most ancestral node dates for the clusters of that variant. For the third analysis, simulations were performed by randomly throwing down different numbers of transition events (40, 80, 120, 160, and 200) along the branches of the Alpha variant clades (clusters) introduced into the study area, with a maximum of one transition event per phylogeny branch.

## Acknowledgments

We thank Joan Cangiarella and Dafna Bar Sagi at NYU Langone Health for supporting the genomic surveillance efforts with institutional funds. We are also grateful to Denise Kühnert for their constructive comments on a previous version of the manuscript. The NYU Genome Technology Center is partially supported by Cancer Center Support (grant n˚P30CA016087). Computational resources have been provided by the *Consortium des Équipements de Calcul Intensif* (CÉCI), funded by the *Fonds de la Recherche Scientifique de Belgique* (F.R.S.-FNRS) under grant n˚2.5020.11 and by the Walloon Region.

## Author Contributions

**Conceptualization:** Simon Dellicour, Philippe Lemey, Guy Baele, Ralf Duerr.

**Data curation:** Simon Dellicour, Christian Marier.

**Formal analysis:** Simon Dellicour, Samuel L. Hong, Verity Hill, Dacia Dimartino, Christian Marier, Paul Zappile.

**Funding acquisition:** Simon Dellicour, Adriana Heguy.

**Investigation:** Simon Dellicour, Samuel L. Hong, Guy Baele, Ralf Duerr.

**Methodology:** Simon Dellicour.

**Project administration:** Simon Dellicour, Ralf Duerr, Adriana Heguy.

**Resources:** Simon Dellicour, Dacia Dimartino, Christian Marier, Paul Zappile, Gordon W. Harkins, Philippe Lemey, Guy Baele, Adriana Heguy.

**Software:** Simon Dellicour, Verity Hill, Philippe Lemey, Guy Baele.

**Supervision:** Simon Dellicour.

**Validation:** Simon Dellicour, Samuel L. Hong, Philippe Lemey, Ralf Duerr, Adriana Heguy.

**Visualization:** Simon Dellicour.

**Writing – original draft:** Simon Dellicour.

**Writing – review & editing:** Simon Dellicour, Samuel L. Hong, Verity Hill, Dacia Dimartino, Christian Marier, Paul Zappile, Gordon W. Harkins, Philippe Lemey, Guy Baele, Ralf Duerr, Adriana Heguy.

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
