## [Decision Letter · Decision Letter 0]

10 Jan 2023

Dear Dellicour,

Thank you very much for submitting your manuscript "Variant-specific introduction and dispersal dynamics of SARS-CoV-2 in New York City – from Alpha to Omicron" for consideration at PLOS Pathogens. As with all papers reviewed by the journal, your manuscript was reviewed by members of the editorial board and by several independent reviewers. In light of the reviews (below this email), we would like to invite the resubmission of a significantly-revised version that takes into account the reviewers' comments.

Two reviewers have seen the paper, both agree there is merit here and that the method and analyses are worthy of publication. Yet both raise some concerned both by possible biases in sequencing data, and by tree topology uncertainty. Notably there is also some reliance on NextStrain sampling - how would this impact the results. I would be happy to see a revised version that addresses all the below comments.

We cannot make any decision about publication until we have seen the revised manuscript and your response to the reviewers' comments. Your revised manuscript is also likely to be sent to reviewers for further evaluation.

Sincerely,

Adi Stern

Academic Editor

PLOS Pathogens

Debra Bessen

Section Editor

PLOS Pathogens

Kasturi Haldar

Editor-in-Chief

PLOS Pathogens

orcid.org/0000-0001-5065-158X

Michael Malim

Editor-in-Chief

PLOS Pathogens

orcid.org/0000-0002-7699-2064

Two reviewers have seen the paper, both agree there is merit here and that the method and analyses are worthy of publication. Yet both raise some concerned both by possible biases in sequencing data, and by tree topology uncertainty. Notably there is also some reliance on NextStrain sampling - how would this impact the results. I would be happy to see a revised version that addresses all the below comments.

Reviewer's Responses to Questions

**Part I - Summary**

Reviewer #1: In this study the authors propose a set of metrics to compare the introduction and dispersal dynamics of the main VOI/VOCs that spread across the NYC area. The metrics are based on a phylogeographic approach, including both discrete and continuous phylogeographic implementations. They apply these metrics to large sets of SARS-CoV-2 genomes and find that interesting differences in the introduction and dispersal dynamics of Iota, Alpha, Delta and Omicron.

Reviewer #2: This generally well written and well performed study by Delicour and colleagues develops and implements a novel comparative framework for phylogeographic analysis of SARS-CoV-2 variants of concern. The authors apply their framework to understand the dynamics of different variants of concern in the New York City area. What makes this paper a novel significant contribution is the analytical framework. This framework is likely to be of great interest to others. A weakness of the study is the dependence on single ML topologies for each variant. Inherent in SARS-CoV-2 phylogenies is substantial phylogenetic uncertainty, it would be prudent to condition analyses on a modest distribution of trees. The figures are useful and appropriate display items to illustrate the main results and caveats are acknowledged especially those with regard to the focal NYC dataset.

**Part II – Major Issues: Key Experiments Required for Acceptance**

Reviewer #1: The introduced metric seem intuitive and the reported results are quite believable, however, the authors are introducing their new metrics p1, p2 and p3 here without having tested them. The performance of these metrics under a small range of simulated scenarios needs to be evaluated. For that it should be sufficient to simulate a small set of introduction and dispersal scenarios, from which sequence data can be generated which is then analysed in the same fashion as presented here. This could also be used to (i) test how much data is required to robustly infer the introduction and dispersal dynamics and (ii) test how sensitive the metrics are to biased sampling.

I compliment the authors for making all the analysis files accessible through their github repository. However, it seems that none of the entire-lineage analyses converged (e.g. Delta_Thorney.log), which is worrisome. This makes me doubt the reliability of the resulting phylogenies, which are the basis for all of the following analyses?

Reviewer #2: The only major issue here is that the authors should consider accounting for inherent phylogenetic uncertainty by inferring even a modest distribution of trees (as other authors have done for phylogeographic studies of SARS-CoV-2) and then running their downstream phylogeographic analyses on this set of trees rather than relying upon a single topology.

**Part III – Minor Issues: Editorial and Data Presentation Modifications**

Reviewer #1: Fig2: Why 80% HPD instead of the more established 95%?

ll.160f "This probability": which probability is meant here?

Reviewer #2: Line 25: “While the main research focus centred on the ability...” reword to be clear the authors mean the main focus of research more generally as worded it could be related to this effort, or the authors own main focus.

“While the main focus of research has centred on the ability…”

Line 232: Change “This index has tended to be decrease over the epidemic waves…” to “This index has tended to decrease over the epidemic waves…”

PLOS authors have the option to publish the peer review history of their article (what does this mean?). If published, this will include your full peer review and any attached files.

Reviewer #1: No

Reviewer #2: No
---

## [Decision Letter · Decision Letter 1]

9 Apr 2023

Dear Dellicour,

We are pleased to inform you that your manuscript 'Variant-specific introduction and dispersal dynamics of SARS-CoV-2 in New York City – from Alpha to Omicron' has been provisionally accepted for publication in PLOS Pathogens.

Best regards,

Adi Stern

Academic Editor

PLOS Pathogens

Debra Bessen

Section Editor

PLOS Pathogens

Kasturi Haldar

Editor-in-Chief

PLOS Pathogens

orcid.org/0000-0001-5065-158X

Michael Malim

Editor-in-Chief

PLOS Pathogens

orcid.org/0000-0002-7699-2064

The authors have done a good job addressing all the comments. Looking forward to seeing it out!

Reviewer Comments (if any, and for reference):

Reviewer's Responses to Questions

**Part I - Summary**

Reviewer #1: The authors have addressed my concerns satisfactorily.

**Part II – Major Issues: Key Experiments Required for Acceptance**

Reviewer #1: (No Response)

**Part III – Minor Issues: Editorial and Data Presentation Modifications**

Reviewer #1: (No Response)

PLOS authors have the option to publish the peer review history of their article (what does this mean?). If published, this will include your full peer review and any attached files.

Reviewer #1: **Yes: **Denise Kühnert

---

## [Editor Report · Acceptance letter]

17 Apr 2023

Dear Dellicour,

We are delighted to inform you that your manuscript, "Variant-specific introduction and dispersal dynamics of SARS-CoV-2 in New York City – from Alpha to Omicron," has been formally accepted for publication in PLOS Pathogens.

Best regards,

Kasturi Haldar

Editor-in-Chief

PLOS Pathogens

orcid.org/0000-0001-5065-158X

Michael Malim

Editor-in-Chief

PLOS Pathogens

orcid.org/0000-0002-7699-2064